# Structure and Fungicidal Activity of Secondary Metabolites Isolated from *Trichoderma hamatum* b-3

**DOI:** 10.3390/jof10110755

**Published:** 2024-10-31

**Authors:** Li Huang, Qiang Bian, Mengdan Liu, Yiwen Hu, Lijuan Chen, Yucheng Gu, Qiwei Zu, Guangzhi Wang, Dale Guo

**Affiliations:** 1State Key Laboratory of Southwestern Chinese Medicine Resource, School of Pharmacy, Chengdu University of Traditional Chinese Medicine, Chengdu 611137, China; 2National Pesticide Engineering Research Center (Tianjin), College of Chemistry, Nankai University, Tianjin 300071, China; bianqiang@nankai.edu.cn; 3School of Basic Medical Sciences, Chengdu University of Traditional Chinese Medicine, Chengdu 611137, China; 4Syngenta Jealott’s Hill International Research Centre, Syngenta, Berkshire RG42 6EY, UK; 5Department of Biochemistry, College of Art & Science, Baylor University, Waco, TX 76706, USA

**Keywords:** *Trichoderma*, diterpene, sesquiterpene, fungicidal activity

## Abstract

Two new harziane diterpenes (**1**–**2**), five undescribed cyclonerane sesquiterpenes (**3**–**7**), and three known compounds, 11-cycloneren-3, 7, 10-triol (**8**), harziandione (**9**), and dehydroacetic acid (**10**), were isolated from *Trichoderma hamatum* b-3. Their structures were elucidated via comprehensive inspection of spectral evidence in HRESIMS and 1D and 2D NMR, and the absolute configuration of **1**–**8** was confirmed by NMR, ECD calculation, as well as Mosher’s method. In vitro fungicidal activity showed that some compounds showed great inhibitory activity against pathogenic fungi, including *Fusarium graminearum*, *Sclerotinia sclerotiorum*, *Botrytis cinerea*, and *Rhizoctonia solani*, among which compound **10** showed 100% inhibition of *S. sclerotiorum* and *B. cinerea*. The in vivo activity test showed that compound **10** was 65.8% effective against *B. cinerea* and compound **10** can be used as a lead compound for the development of biopesticides that inhibit *B. cinerea*. This study elucidated the bioactivity of secondary metabolites of *T. hamatum* and indicated the direction for the subsequent development of the biological control activity of *T. hamatum*.

## 1. Introduction

The continuous growth of the global population exerts tremendous pressure on agricultural production systems. Concurrently, the food and economic losses caused by plant pathogens in agriculture have raised serious concerns [1,2]. Chemical control can mitigate plant diseases and enhance grain yield, making it one of the most effective methods for prevention and control. However, the frequent use of chemical pesticides can result in pesticide residues in crops, contaminate soil and water sources, as well as pose risks to human health [3,4]. The European Commission has recently proposed a ‘Green Deal’ aimed at reducing the use of chemical pesticides by 50 percent by the year 2030 [5]. Biological control entails managing plant diseases through the use of microorganisms and their metabolites, which act on pathogens through mechanisms such as parasitism, the production of antibiotics or secondary metabolites, competition for living space, and the induction of plant resistance [6]. Compared to chemical pesticides, biological control is likely to be less toxic to crops, more selective for target pathogens, degrades rapidly, and is less prone to the development of resistance [7]. As green agriculture evolves, there is an urgent need to identify new alternatives to chemical control, with biological control representing a viable option.

*Trichoderma* strains have been effectively employed for the biological control of plant diseases, owing to their diverse biocontrol mechanisms, which include competition, antibiosis, induced systemic resistance, and mycoparasitism [8,9,10,11]. Additionally, a range of biocontrol formulations based on *Trichoderma* have been successfully developed. Antibiosis is a primary biological control mechanism employed by *Trichoderma*, which involves the production of secondary metabolites that inhibit the growth of pathogenic fungi [12,13]. *Trichoderma* strains are known to produce a diverse array of natural products, including epipolythiodioxopiperazines [14], peptaibols [15], pyrones [16], butenolides [17], koninginins [18], steroids [19], lactones [20], and trichothecenes [21,22]. 6-Pentyl-2H-pyran-2-one (6-pp) is a volatile metabolite produced by *Trichoderma*, which not only promotes plant growth and development but also effectively inhibits the growth of pathogenic bacteria, including *Phytophthora capsica*, *Fusarium oxysporum*, *Rhizoctonia solani*, and *Sclerotium rolfsii* [23]. Our research team has been dedicated to the study of fungal secondary metabolites with bioactive properties over an extended period, resulting in the isolation of several previously undescribed compounds from fungal strains, such as *Chaetomium elatum* [24] and *Nigrospora sphaerica* [25]. In this study, we investigated the secondary metabolites of *Trichoderma hamatum*, leading to the isolation of two new harziane diterpenes (**1**–**2**) and five cyclonerane sesquiterpenes (**3**–**7**). The structures of these compounds were determined by HRESIMS, NMR, ECD, and Mosher’s method [26], combined with chemical calculation. Compounds **1**–**10** were tested for their in vitro fungicidal activity against six agricultural pathogenic fungi. Compound **10** was then tested for in vivo fungicidal activities against some pathogenic fungi. Herein, we report the details of the isolation, elucidation, and fungicidal activity of the above-mentioned secondary metabolites.

## 2. Materials and Methods

### 2.1. General Experimental Procedures

Optical rotation was measured on a Perkin Elmer 241 Polarimeter (Perkin Elmer, Inc., Waltham, MA, USA). UV, CD, and ICD spectra were measured on a Chirascan circular dichroism (Applied Photophysics Ltd., Leatherhead, UK). The IR spectrum was acquired from an Agilent Cary 600 FT-IR infrared spectrometer (Agilent Technologies, Santa Clara, CA, USA). The mass spectrum was obtained via ultra-performance liquid chromatography coupled with a Q Exactive quadrupole-electrostatic field orbital trap high-resolution mass spectrometer (Thermo Fisher Scientific, Bremen, Germany). NMR spectra were measured with a Bruker Ascend, 600 MHz (Burker, Karlsruhe, Germany), with TMS as an internal standard. Fractionation was conducted on a column chromatography silica gel (200–300 mesh). Samples were fractionated by a dynamic axial compression column (Hanbon Sci. & Tech, Huaian, China). Purification was performed with a NP7000 preparative high-performance liquid chromatograph (Hanbon Sci. & Tech, Huaian, China) with a kromasil C18 5 μm semi-preparative column (10 × 250 mm). Methanol, acetone, ethyl acetate (analytically pure), and dimethyl sulfoxide (chromatographic pure) were purchased from Chengdu Kelong Chemical Co., Ltd. (Chengdu, China). Dichloromethane (analytically pure), methanol (chromatography preparative pure), and acetonitrile (chromatography preparative pure) were purchased from Chengdu Jinshan Chemical Reagent Co., Ltd. (Chengdu, China). Deuterated chloroform was purchased from Adamas Reagent Co., Ltd. (Shanghai, China). N, N-dimethylformamide (analytically pure) was purchased from Tianjin Fuyu Fine Chemical Co. Ltd. (Tianjin, China). Pyrimethanil and dimethachlon (analytically pure) were purchased from Alta Scientific Co., Ltd. (Tianjin, China). 

### 2.2. Fungal Material and Fermentation

The tissue isolation method was employed to isolate and purify endophytic fungi from fresh *Bergenia purpurascens* plants collected from Emei Mountain, resulting in the acquisition of strain b-3. Five days after incubation on PDA medium, the mycelium of strain b-3 covered a diameter of approximately 9 cm, exhibiting white mycelium with radial edges (Appendix A). Initially, the mycelium appeared white, but it later produced green conidia that spread outward from the center. The conidia were either round or ellipsoid (Appendix A), and the conidial peduncle was short, expanding in the middle and tapering at the top (Appendix A). Based on these characteristics, the strain was initially identified as *Trichoderma hamatum*.

After inoculating b-3 into PDA medium for 5 days, the fungal genomic DNA was extracted using the E.Z.N.A Fungal DNA Mini Kit (Beijing Solarbio Science & Technology Co., Ltd., Beijing, China). The b-3 strain was amplified by PCR utilizing the fungal universal primers, ITS1 (5’-TCCGTAGGTGAACCTGCGG-3’) and ITS4 (5’-TCCTCCGCTTATTGATATGC-3’). The rDNA-ITS region of b-3 was specifically amplified via PCR with the following reaction system (25 μL): 12.5 μL of 2 × Taq PCR Master Mix, 1 μL each of ITS1 and ITS4 primers (10 μmol/L), 1 μL of DNA template, and 9.5 μL of ddH_2_O. The PCR reaction conditions were set as follows: an initial denaturation at 95 °C for 3 min, followed by 30 cycles of denaturation at 95 °C for 30 s, annealing at 54 °C for 45 s, and extension at 72 °C for 60 s, concluding with a final extension at 72 °C for 10 min. After the PCR reaction was completed, the products were analyzed by electrophoresis on a 1.0% agarose gel. The strain was identified as *T. hamatum* (GenBank Accession No. OR553890 and Sequence No. NR134371.1) by Shanghai Sangong Bioengineering Co., Ltd. through morphological analysis and BLAST comparison.

Fungus No. 2 medium (maltose 20 g, monosodium glutamate 10 g, dextrose 10 g, yeast paste 3 g, corn syrup 1 g, mannitol 20 g, KH_2_PO_4_ 0.5 g, MgSO_4_·7H_2_O 0.3 g, CaCO_3_ 20 g, 5-azacytidine 10 mg, and distilled water 1 L) was used for the cultivation of *T. hamatum* b-3 [27]. The experimental strains were inoculated onto potato dextrose agar (PDA) medium for a duration of 5 days. Subsequently, five pieces (0.5 × 0.5 cm^2^) of mycelial agar plugs were transferred into sterilized and cooled Fungi No. 2 liquid medium contained in 500 mL conical flasks, with 200 mL of medium per flask. These flasks were then placed in a constant-temperature shaker set at 28 °C and 120 rpm/min for 7 days to obtain the *T. hamatum* seed cultures. Following this, the seed cultures were inoculated into the sterilized Fungus 2 medium for expansion cultures, with 2.5 mL of seed cultures added into each 100 mL of medium. The inoculated medium was cultured in a constant-temperature shaker at 28 °C and 120 rpm/min for 14 days to produce the fermentation broth of *T. hamatum* b-3, resulting in a total culture volume of 564.4 L.

### 2.3. Extraction and Isolation

The cultured fermentation broth was centrifuged once in a high-speed centrifuge at 4000 r/min to separate the fermentation broth. The broth was then treated with an equal volume of ethyl acetate for 24 h. Subsequently, the extraction process was repeated three times using an equal volume of ethyl acetate. The ethyl acetate layer was collected and concentrated under reduced pressure, yielding 126 g of ethyl acetate extract.

The ethyl acetate extract was separated by silica gel CC, eluted with petroleum ether and acetone (30:1~1:1, *v*/*v*), yielding seven fractions (Fr.1~Fr.7). Fr.1 was separated through column chromatography on dynamic axial compression column (MeOH/H_2_O, 90:10) and semi-pHPLC (MeOH/H_2_O, 68:32) to obtain **9** (11.5, *t_R_* 20.3 min). Fr.2 was separated via RP-C18 CC (MeOH/H_2_O, 95:5) to afford three fractions (Fr.2.1~Fr.2.3). Fr.2.1 was then separated through semi-pHPLC (MeOH/H_2_O, 50:50 to 100:0) to obtain eleven fractions (Fr.2.1.1~Fr.2.1.11), and Fr.2.1.5 was then separated through semi-pHPLC (MeOH/H_2_O, 40:60) to yield **10** (87 mg, *t_R_* 25.8 min). Fr.2.1.6 was subjected to silica gel CC and semi-pHPLC (MeOH/H_2_O, 70:30) to yield **1** (1.8 mg, *t_R_* 28.0 min). Fr.2.1.8 was separated via Sephadex LH-20 CC (CH_2_Cl_2_/MeOH, 70:30) and purified by semi-pHPLC (MeCN/H_2_O, 20:80) to produce **8** (13.5 mg, *t_R_* 30.3 min) and semi-pHPLC (MeCN/H_2_O, 24:76) to produce **4** (1.8 mg, *t_R_* 19.7 min) and **5** (9.3 mg, *t_R_* 26.6 min). Fr.3 was separated through dynamic axial compression column chromatography (MeOH/H_2_O, 50:50 to 50:100) to yield six fractions (Fr.3.1~Fr.3.6), and Fr.3.4 was further purified by CC on Sephadex LH-20 (MeOH) as well as semi-pHPLC (MeCN/H_2_O, 57:43) to afford **3** (1.3 mg, *t_R_* 8.3 min). Fr.3.4.1.10 was further purified by semi-pHPLC (MeCN/H_2_O, 35:65) to obtain **6** (1.4 mg, *t_R_* 35.4 min). Fr.5 was separated via dynamic axial compression column chromatography (MeOH/H_2_O, 60:40 to 100:0) to yield five fractions (Fr.5.1~Fr.5.5), and Fr.5.2 was then purified by CC on Sephadex LH-20 (CH_2_Cl_2_/MeOH, 70:30) and semi-pHPLC (MeCN/H_2_O, 40:60) to yield **7** (5.7 mg, *t_R_* 19.4 min).

Compound **1**: Colorless oil; ${\left[\alpha \right]}_{D}^{20}$ = +81.8 (*c* = 0.011, MeOH); UV (MeOH) *λ*_max_ 248 (0.58); IR (KBr): 3449, 2920, 2851, 1740, 1384, 1234, 1026 cm^−1^; CD (c 2.23 mM, MeOH) λ_max_ (Δ*ε*) 238 (−1.21), 295 (0.53), 342 (0.67) nm; HRESIMS *m/z* 359.2212 [M+H]^+^ (calcd. for C_22_H_31_O_4_^+^, 359.2217). For ^1^H-NMR and ^13^C-NMR data, see Table 1 and Table 2.

Compound **2**: Colorless oil; ${\left[\alpha \right]}_{D}^{20}$ = +7.1 (*c =* 0.028, MeOH); UV (MeOH) *λ*_max_ 255 (0.48); IR (KBr): 3439, 2936, 1730, 1643, 1386, 1199 cm^−1^; CD (c 2.51 mM, MeOH) λ_max_ (Δ*ε*) 222 (−0.31), 251 (−0.73), 299 (0.11), 343 (0.53) nm; HRESIMS *m/z* 319.2268 [M+H]^+^ (calcd. for C_20_H_31_O_3_^+^, 319.2268). For ^1^H-NMR and ^13^C-NMR data, see Table 1 and Table 2.

Compound **3**: Colorless oil; ${\left[\alpha \right]}_{D}^{20}$ = −8.0 (*c =* 0.025, MeOH); UV (MeOH) *λ*_max_ 209 (4.32); IR (KBr): 3436, 2958, 2926, 2850, 1666, 1622, 1386 cm^−1^; CD (c 3.05 mM, MeOH) λ_max_ (Δ*ε*) 224 (0.33), 321 (−0.12) nm; HRESIMS *m/z* 263.1624 [M+Na]^+^ (calcd. for C_14_H_24_NaO_3_^+^, 263.1618). For ^1^H-NMR and ^13^C-NMR data, see Table 3 and Table 4.

Compound **4**: Colorless oil; ${\left[\alpha \right]}_{D}^{20}$ = −24.0 (*c =* 0.05, MeOH); IR (KBr): 3424, 2962, 2875, 1716, 1647, 1455, 1377, 918 cm^−1^; HRESIMS *m/z* 279.1940 [M+Na]^+^ (calcd. for C_15_H_28_NaO_3_^+^, 279.1931). For ^1^H-NMR and ^13^C-NMR data, see Table 3 and Table 4.

Compound **5**: Colorless oil; ${\left[\alpha \right]}_{D}^{20}$ = −19.7 (*c =* 0.20, MeOH); UV (MeOH) *λ*_max_ 213 (0.49); IR (KBr): 3402, 2966, 2931, 1458, 1377, 1156, 920 cm^−1^; CD (c 2.94 mM, MeOH) λ_max_ (Δ*ε*) 216 (−0.05) nm; HRESIMS *m/z* 271.1910 [M-H]^−^ (calcd. for C_15_H_27_O_4_^−^, 271.1914). For ^1^H-NMR and ^13^C-NMR data, see Table 3 and Table 4.

Compound **6**: Colorless oil; ${\left[\alpha \right]}_{D}^{20}$ = −3.2 (*c =* 0.06, MeOH); IR (KBr): 3453, 2964, 2937, 1737, 1459, 1383, 1211, 1169 cm^−1^; CD (c 1.62 mM, MeOH) λ_max_ (Δ*ε*) 205 (−0.21), 243 (−0.04), 307 (0.3) nm; HRESIMS *m/z* 393.2249 [M+Na]^+^ (calcd. for C_20_H_34_NaO_6_^+^, 393.2248). For ^1^H-NMR and ^13^C-NMR data, see Table 1 and Table 2.

Compound **7**: Colorless oil; ${\left[\alpha \right]}_{D}^{20}$ = −4.9 (*c =* 0.14, MeOH); IR (KBr): 3456, 2968, 2936, 1738, 1438, 1379, 1217, 1163 cm^−1^; CD (c 2.16 mM, MeOH) λ_max_ (Δ*ε*) 221 (0.05) nm; HRESIMS *m/z* 393.2252 [M+Na]^+^ (calcd. for C_20_H_34_NaO_6_^+^, 393.2248). For ^1^H-NMR and ^13^C-NMR data, see Table 1 and Table 2.

Compound **8**: Colorless oil; ${\left[\alpha \right]}_{D}^{20}$ = −26.8 (*c =* 0.28, MeOH); IR (KBr): 3408, 2960, 2873, 1719, 1647, 1455, 1372, 921 cm^−1^; HRESIMS *m/z* 279.1941 [M+Na]^+^ (calcd. for C_15_H_28_NaO_3_^+^, 279.1931). For ^1^H-NMR and ^13^C-NMR data, see Table 3 and Table 4.

### 2.4. NMR and ECD Calculation Methods 

The chemical calculations of compounds were conducted using Gaussian 16 ^1^. Initially, a conformational analysis was performed with Conflex 8 (CONFLEX Corporation, Tokyo, Japan) to generate conformations via Boltzmann Jump [28]. All geometric configurations with relative energies between 0 and 5.0 kcal/mol were optimized at the B3LYP/6-31G level in the gas phase, as well as at the ωB97XD/DGDZVP level in methanol. Room-temperature equilibrium populations were determined based on the Boltzmann distribution law [29]. Shielding tensor calculations were conducted at the PCM/mPW1PW91/6-31+G (d, p) level (with Boltzmann distribution ≥ 1%) employing the GIAO method [30]. The isotropic values of TMS were calculated at the same level and used as a reference. The DP4+ parameters were computed using the Excel file provided by Sarotti [31]. ECD calculations were performed using TD-DFT at the CAM-B3LYP/DGDZVP level in methanol. The ECD spectra were generated by considering the Boltzmann distribution of each geometric conformation. Subsequently, SpecDis 1.71 was utilized to combine the individual CD spectra with a Boltzmann statistical weighting, resulting in a Gaussian curve (σ = 0.16–0.4 eV), which was then compared with experimental data.

### 2.5. Fungicidal Activity Assay of Compounds **1**–**10** In Vitro

Pathogenic fungi, including *Alternaria solani* that cause tomato early blight, *Fusarium graminearum* that cause wheat scab, *Phytophthora capsici* that cause pepper phytophthora blight, *Sclerotinia sclerotiorum* that cause rape sclerotinia stem rot, *Botrytis cinerea* that cause grey mold in cucumbers and tomatoes, and *Rhizoctonia solani* that cause rice blight, were inoculated onto petri dishes containing a compound solution at a concentration of 50 μg/mL. The dishes were then incubated in a biochemical incubator at 25 °C in the dark. The assessment of bactericidal activity was conducted following a three-day incubation period, with each experimental group replicated three times. The control group was treated with sterile water. The results of the activity assessment were quantified on a percentage scale ranging from 0 to 100, where 0 indicates no activity and 100 signifies complete eradication [32]:$$Control\;effect\;(\%)=\frac{blank\;colony\;diameter-colony\;diameter\;after\;liquid\;treatment}{blank\;colony\;diameter-4}\times 100$$

### 2.6. Fungicidal Activity Assay of Compound **10** In Vivo

The commercial fungicides pyrimethanil and dimethachlon were utilized as positive controls (PC) in this study. The compound **10** and the control were dissolved in N, N-dimethylformamide and subsequently diluted to a concentration of 200 µg/mL, with water as blank control (CK). Cucumber plants were then sprayed with these solutions and allowed to air-dry for approximately 2 h. Following this drying period, the undersurfaces of the treated cucumber leaves were sprayed with a pathogen spore suspension containing approximately 1 × 10^4^ spores/mL. The plants were then placed in an incubator maintained at 20 °C with humidity levels exceeding 90% for a 5-day infection period. After this initial incubation, the plants were transferred to a greenhouse for an additional 5 days before being assessed for disease control scores [32].

## 3. Results

### 3.1. Structural Identification of Compounds

Compound **1** was isolated as a colorless oil, with its molecular formula determined to be C_22_H_30_O_4_ by HRESIMS, revealing eight degrees of unsaturation. The FT-IR absorption bands indicated the presence of hydroxyl (3449 cm^−1^), methyl (2920 and 2851 cm^−1^), and carbonyl (1740 cm^−1^) groups. The ^1^H-NMR, ^13^C-NMR (Table 1 and Table 2), and HSQC spectra revealed the presence of three carbonyl groups (*δ*_C_: 214.2 (C-3), 196.5 (C-11), and 170.9 (C-21)), a set of conjugated double bonds (*δ*_C_: 153.0 (C-10) and 141.1 (C-9)), six aliphatic methylene groups (*δ*_H_: 5.12 (H-20a), 4.76 (H-20b), 2.90 (H-4a), 2.66 (H-12a), 2.51 (H-12b), 2.29 (H-8), 2.09 (H-4b), 2.05 (H-15a), 1.97 (H-7a), 1.54 (H-15b), and 1.40 (H-7b); *δ*_C_: 63.3 (C-20), 60.2 (C-12), 42.7 (C-4), 30.3 (C-7), 26.7 (C-15), and 23.8 (C-8)), three hypomethyl groups (*δ*_H_: 2.90 (H-5), 2.52 (H-14), and 2.29 (H-2); *δ*_C_: 59.4 (C-2), 52.9 (C-14), and 30.1 (C-5)), five methyl groups (*δ*_H_: 2.10 (H-22), 1.54 (H-19), 1.13 (H-18), 1.01 (H-17), and 1.00 (H-16); *δ*_C_: 25.2 (C-16), 23.4 (C-17), 21.1 (C-18), 21.0 (C-22), and 20.6 (C-19)), and three quaternary carbons (*δ*_C_: 51.8 (C-6), 49.7 (C-1), and 40.4 (C-13)). The ^1^H-^1^H COSY (Figure 1) correlations of H-2/H-15/H-14, H-4/H-5/H-18, and H-7/H-8, along with HMBC correlations of H-2, H-4, H-15/C-3; H-18/C-4, C-5, C-6; H-16, H-17/C-1, C-2, C-6; H-7/C-5; H-19/C-10, C-12, C-13, C-14; H-20/C-8, C-9, C-10, C-21; and of H-22/C-21, elucidated the planar structure of **1**, as shown in Figure 2.

The relative configuration of compound **1** was determined to be 2S*, 5R*, 6R*, 13S*, 14S* based on the NOESY correlations of H-16/H-14, H-2, and H-5/H-19. The absolute configuration of **1** was confirmed to be 2*S*, 5*R*, 6*R*, 13*S*, 14*S* by subsequent ECD calculations (Figure 3).

Compound **2** was purified as a colorless oil, and its molecular formula was deduced to be C_20_H_30_O_3_ by (+)-HRESIMS *m*/*z* 319.2268 [M+H]^+^ (calcd. for C_20_H_31_O_3_^+^, 319.2268), implying six degrees of unsaturation. The IR spectrum of **2** showed absorption bands for hydroxyl (3439 cm^−1^), methyl (2936 cm^−1^), and carbonyl (1730 cm^−1^). The ^1^H-NMR, ^13^C-NMR (Table 1 and Table 2), and HSQC spectra indicated that compound **2** is a harziane diterpenes-type compound, similar to compound **1**, with the difference that the substituents of compound **2** at C-3 and C-20 are both hydroxyl groups. The entire structure was confirmed by the ^1^H–^1^H COSY correlations of H-14/H-15/H-2/H-3/H-4/H-5/H-18 and H-7/H-8, and the HMBC correlations of H-7/C-5; H-12/C-11; H-16, H-17/C-1, C-2, C-6; H-18/C-4, C-5, C-6; H-19/C-10, C-12, C-13, C-14, as well as H-20/C-8, C-9, C-10. The relative configuration of compound **2** was assigned to be 2*S**, 3*S**, 5*R**, 6*R**, 13*S**,14*S** by the NOESY correlation of H-15b/H-19, H-3 and H-19/H-5. The experimental and calculated ECD spectra did match well, suggesting 2*S*, 3*S*, 5*R*, 6*R*, 13*S*, 14*S* was the correct absolute configuration (Figure 3).

Compound **3** was isolated as a colorless oil, and its molecular formula was deduced to be C_14_H_24_O_3_ by (+)-HRESIMS *m*/*z* 263.1624 [M+Na]^+^ (calcd. for C_14_H_24_NaO_3_^+^, 263.1618), implying three degrees of unsaturation. The FT-IR absorption band suggested the presence of a hydroxy group (3436 cm^−1^), methyl group (2958 cm^−1^, 2926 cm^−1^, and 2850 cm^−1^), olefinic group (1622 cm^−1^), and carbonyl group (1666 cm^−1^). The ^1^H-NMR, ^13^C-NMR (Table 3 and Table 4), and HSQC spectra indicated the presence of one carbonyl signal (*δ*_C_: 198.5 (C-11)), a group of conjugated olefin signals (*δ*_H_: 6.88 (H-9) and 6.13 (H-10); *δ*_C_: 144.1 (C-9) and 134.2 (C-10)), three aliphatic methylene signals (*δ*_H_: 2.45 (H-8a), 2.36 (H-8b), 1.91 (H-5a), 1.72 (H-4a), and 1.58 (H-4b, 5b); *δ*_C_: 43.8 (C-8), 40.4 (C-4), and 24.6 (C-5)), two hypomethyl signals (*δ*_H_: 1.86 (H-6) and 1.62 (H-2); *δ*_C_: 54.9 (C-6) and 44.6 (C-2)), two oxidized quaternary carbon signals (*δ*_C_: 81.4 (C-3) and 75.0 (C-7)), and four methyl signals (*δ*_H_: 2.27 (H-12), 1.27 (H-13), 1.19 (H-14), and 1.06 (H-1); *δ_C_*: 27.2 (C-12), 26.2 (C-13), 25.9 (C-14), and 14.6 (C-1)). The ^1^H–^1^H COSY correlations of H-1/H-2/H-6/H-5/H-4 and H-8/H-9/H-10, as well as HMBC correlations of H-1/C-3, H-13/C-2, C-3, C-4; H-14/C-6, C-7, C-8; H-10/C-11, and H-12/C-10, C-11 generated the planar structure of **3**. The configuration of the double bond at C-9 was assigned as trans by the NOESY correlation signal of H-8/H-10, and the large coupling constant between H-9 and H-10. The NOESY correlations of H-2/H-13 and H-1/H-6 indicated the relative configuration of C-2, C-3, C-6. 

The absolute configuration of C-7 could not be determined by ECD. To solve this difficult stereoscopic problem, the NMR data of **3**a (2*S**, 3*R**, 6*R**, 7*R**) and **3**b (2*S**, 3*R**, 6*R**, 7*S**) were further calculated at the PCM/mPW1PW91/6–31+G (d, p) level using the GIAO’s method [29]. The calculated ^13^C-NMR chemical shifts of **3**a showed a better agreement with the experimental values of compound **3**, with a higher correlation coefficient (R^2^ for **3**a: 0.9958; R^2^ for **3**b: 0.9956; Appendix A). In addition, DP4+ probability analysis [30] based on both ^1^H- and ^13^C-NMR data predicated **3**a as the correct relative structure, with 96.49% probability (Appendix A). The experimental and calculated ECD spectra for 2*S*, 3*R*, 6*R*, 7*R* in MeOH did match well, indicating that 2*S*, 3*R*, 6*R*, 7*R* was the correct absolute configuration (Figure 3).

Compound **4** was isolated as a colorless oil, and its molecular formula was determined as C_15_H_28_O_3_ by (+)-HRESIMS *m*/*z* 279.1940 [M+Na]^+^ (calcd. for C_15_H_28_NaO_3_^+^, 279.1931), implying two degrees of unsaturation. The FT-IR absorption band indicated that compound **4** contained a hydroxyl group (3424 cm^−1^), methyl group (2962 and 2875 cm^−1^), and olefinic group (1647 cm^−1^). The planar structure of **4** was deduced to be the same as that of 11-cycloneren-3, 7, 10-triol [33] via comparison of its ^1^H- and ^13^C-NMR data (Table 3 and Table 4) with those for **4**, which was supported by HMBC correlations and ^1^H–^1^H COSY correlations (Figure 1).

The relative configurations of the cyclopentane in compound **4** were determined to be *2S**, *3R**, *6R** by the NOESY-related signals of H-1/H-6, as well as H-2/H-13. Since the amount of compound **4** was not sufficient for the Mosher’s reaction, we determined its absolute configuration by calculating NMR and ECD. The relative configuration of compound **4** was further determined to be 2*S**, 3*R**, 6*R**, 7*R**, 10*R** based on the DP4+ probability (100% for 2*S**, 3*R**, 6*R**, 7*R**, 10*R**; Appendix A) analysis of ^1^H-NMR and ^13^C-NMR data. The absolute configuration of compound **4** was determined to be 2*S*, 3*R*, 6*R*, 7*R*, 10*R* based on the comparison of calculated and experimental ECD spectra (Figure 3).

Compound **5** was separated as a colorless oil, with the molecular formula C_15_H_28_O_4_ established by the (-)-HRESIMS data at *m*/*z* 271.1910 [M-H]^−^ (calcd. for C_15_H_27_O_4_^−^ 271.1914), indicating two degrees of unsaturation. The FT-IR absorption band indicated that compound **5** contained a hydroxyl group (3402 cm^−1^) and methyl group (2966 and 2931 cm^−1^). The NMR data of **5** (Table 3 and Table 4) closely resembled those of 9-cycloneren-3, 7, 11-triol [30], with the main difference being that -OH at the C-10 was substituted with -OOH. This was demonstrated by the HRESIMS data and the HMBC correlations of H-10/C-11; H-12/C-10, C-11, H-15. Other HMBC and ^1^H–^1^H COSY correlations confirmed the entire structure of compound **5**. The trans relative configuration of C-9 and C-10 was established according to the large coupling constant of H-9 and H-10 and the NOESY correlations of H-10/H-8 (Figure 4). The relative configuration of **5** was determined to be 2*S**, 3*R**, 6*R** by NOESY cross-peaks of H-2/H-13 and H-1/H-6.

Additionally, the relative configuration of C-7 was deduced as 7*R** by NMR calculations with the DP4+ probability of 100% (Appendix A). The absolute configuration of compound **5** was determined to be 2*S*, 3*R*, 6*R*, 7*R* by comparing the calculated ECD and experimental ECD spectra (Figure 5).

Compound **6** was isolated as a colorless oil, and its molecular formula was deduced to be C_20_H_34_O_6_ by (+)-HRESIMS *m*/*z* 393.2249 [M+Na]^+^ (calcd. for C_20_H_34_NaO_6_^+^, 393.2248), with four degrees of unsaturation. The FT-IR absorption band suggested the presence of a hydroxy group (3453 cm^−1^), methyl group (2964 and 2937 cm^−1^), and carbonyl group (1737 cm^−1^). The ^1^H-NMR, ^13^C-NMR (Table 1 and Table 2), and HSQC spectra indicated the presence of two carbonyl signal (*δ*_C_: 172.9 (C-16) and 172.4 (C-13)), a group of conjugated olefin signals (*δ*_H_: 5.41 (H-10); *δ*_C_: 131.2 (C-10) and 129.9 (C-11)), six aliphatic methylene signals (*δ*_H_: 2.64 (H-14, H-15), 2.19 (H-9a), 2.12 (H-9b), 1.85 (H-5a), 1.68 (H-4a), 1.56 (H-4b), 1.55 (H-5b), and 1.49 (H-8); *δ*_C_: 40.5 (C-4, C-8), 29.3 (C-15), 29.1 (C-14), 24.5 (C-5), and 22.5 (C-9)), an oxidized methylene signal (*δ*_H_: 4.66 (H-12a) and 4.60 (H-12b); *δ*_C_: 63.6 (C-12)), two hypomethyl signals (*δ*_H_: 1.84 (H-6) and 1.60 (H-2); *δ*_C_: 54.6 (C-6) and 44.4 (C-2)), two oxidized quaternary carbon signals (*δ*_C_: 81.4 (C-3) and 74.8 (C-7)), four methyl signals (*δ*_H_: 1.74 (H-20), 1.26 (H-18), 1.15 (H-19), and 1.04 (H-1); *δ*_C_: 26.2 (C-18), 25.0 (C-19), 21.6 (C-20), and 14.7 (C-1)), and a methoxy signal (*δ*_H_: 3.69 (H-17); *δ*_C_: 52.0 (C-17)). 

The ^1^H–^1^H COSY correlations of H-1/H-2/H-6/H-5/H-4, H-8/H-9/H-10, and H-14/H-15, as well as HMBC correlations of H-1/C-3; H-18/C-2, C-3, C-4; H-19/C-6, C-7, C-8; H-10/C-8; H-20/C-10, C-11, C-12; H-12/C-13; H-14/C-13; H-15/C-13, C-16, and H-17/C-16 generated the primary structure of **6**. The configuration of the double bond at C-10 was assigned as cis by the NOESY correlation signal between H-9/H-12, as well as H-10/H-20. The correlations between H-2 and H-18, and H-1 and H-6 suggested that the relative configurations of C-2, C-3 and C-6 were 2*S*, 3*R*, 6*R*.

The relative configuration of C-7 was 7*R**, as shown by the theoretical NMR calculation using GIAO’s method combined with a DP4+ probability analysis (Appendix A). The absolute configuration of **6** was confirmed by the similarity between the calculated ECD curve of 2*S*, 3*R*, 6*R*, 7*R*-**6** and its experimental ECD spectrum (Figure 5). Thus, the structure of **6** is as shown in Figure 2.

Compound **7** was obtained as a colorless oil, and the molecular formula, C_20_H_34_O_6_, was obtained by analysis of (+)-HRESIMS *m*/*z* 393.2252 [M+Na]^+^ (calcd. for C_20_H_34_NaO_6_^+^, 393.2248), suggesting four degrees of unsaturation. The FT-IR absorption band showed the presence of hydroxyl (3456 cm^−1^), methyl (2968 and 2936 cm^−1^), and carbonyl (1738 cm^−1^) groups. Its ^1^H- and ^13^C-NMR (Table 1 and Table 2) as well as HREIMS data closely resembled those of **7**, and the ^1^H–^1^H COSY and HMBC correlations (Figure 1) also suggested that **8** has a similar planar structure to **7**. The difference between compounds **8** and **7** is that the geometry of the double bond at C-10 is trans, which was inferred from the NOESY correlations of H-9/H-20, as well as H-10/H-12. The relative configuration of C-2, C-3, C-6 was determined to be 2*S**, 3*R**, 6*R** based on the correlation signals of H-2/H-18 and H-1/H-6 in the NOESY spectra (Figure 4).

The relative configuration of C-7 was determined by NMR calculations combined with DP4+ probability analysis (100% for 2*S**, 3*R**, 6*R**, 7*R**; Appendix A). By contrasting the experimental and calculated ECD data, the absolute configuration was validated.

Compound **8** was obtained as a colorless oil, and its molecular formula was determined as C_15_H_28_O_3_ by (+)-HRESIMS *m*/*z* 279.1941 [M+Na]^+^ (calcd. for C_15_H_28_NaO_3_^+^, 279.1931), implying two degrees of unsaturation. The FT-IR absorption band indicated that compound **8** contained a hydroxyl group (3408 cm^−1^), methyl group (2960 and 2873 cm^−1^), and olefinic group (1647 cm^−1^). The ^1^H-NMR, ^13^C-NMR (Table 3 and Table 4), and HSQC spectra exhibited the presence of a group of olefinic signals (*δ*_H_: 4.97 (H-12a) and 4.86 (H-12b); *δ*_C_: 147.7 (C-11) and 111.0 (C-12)), four methylene groups (*δ*_H_: 1.87 (H-5a), 1.71 (H-9a), 1.69 (H-4a), 1.61 (H-9b), 1.57 (H-8a, H-4b), 1.56 (H-5b), and 1.50 (H-8b); *δ*_C_: 40.5 (C-4), 35.8 (C-8), 29.1 (C-9), and 24.5 (C-5)), two hypomethyl groups (*δ*_H_: 1.87 (H-6) and 1.59 (H-2); *δ*_C_: 54.8 (C-6) and 44.5 (C-2)), one oxidized hypomethyl group (*δ*_H_: 4.09 (H-10); *δ*_C_: 76.0 (C-10)), two oxidized quaternary carbon signals (*δ*_C_: 81.4 (C-3) and 74.8 (C-7)), and four methyl signals (*δ*_H_: 1.72 (H-15), 1.26 (H-13) 1.16 (H-14), and 1.04 (H-1); *δ*_C_: 26.2 (C-13), 25.2 (C-14), 18.2 (C-15), and 14.6 (C-1)). 

The ^1^H–^1^H COSY correlations of H-1/H-2/H-6/H-5/H-4 and H-8/H-9/H-10, along with the HMBC correlation signals of H-1/C-3, H-13/C-2, C-3, and C-4, H-14/C-6, C-7, and C-8, H-10/C-11, C-12, and H-15/C-10, C-11, and C-12 demonstrated that compound **8** is 11-cycloneren-3, 7, 10-triol [27]. According to previous research [27], the chirality of 11-cycloneren-3, 7, 10-triol at C-10 was undetermined, so the Mosher’s method was used to determine its conformation. Compound **8** was esterified with (*S*)-MTPA-Cl and (*R*)-MTPA-Cl, respectively, to yield Mosher ester derivatives. The *Δδ*_S-R_ of the ^1^H-NMR data of each proton adjacent to C-10 in the (*S*)-MTPA ester and (*R*)-MTPA ester products were compared (Figure 6), and the final absolute configuration of C-10 was determined to be 10*S*. The ECD of compound **8** was calculated, and the calculated ECD curves of 2*S*, 3*R*, 6*R*, 7*R*, 10*S*-**8** were found to be in good agreement with the experimental ECD curves; thus, the absolute configuration of compound **8** was determined.

### 3.2. Fungicidal Activities of Compounds **1**–**10** In Vitro

We tested the fungicidal activity of compounds **1**–**10** against six crop pathogens at a concentration of 50 µg/mL. Fungicidal tests showed that compounds **1**–**10** all showed good inhibition of *S. sclerotiorum* at a concentration of 50 µg/mL. In addition, compounds **3** and **10** showed significant inhibition against *B. cinerea*, with inhibition rates of 81.6*%* and 100%, respectively (Table 5).

### 3.3. Fungicidal Activities of Compound **10** In Vivo

In the in vivo activity test, compound **10** was effective against two pathogens, and it showed significant inhibitory activity against *B. cinerea*, with a control effect of 65.8% (Table 6, Figure 7).

## 4. Discussion

A variety of *Trichoderma* strains have been utilized worldwide as effective biocontrol agents, including *T. harzianum*, *T. hamatum*, *T. asperellum*, *T. atroviride*, *T. koningii*, and *T. viride* [34,35,36,37,38]. One of the mechanisms by which *Trichoderma* exerts biological control is through the production of secondary metabolites [39,40]. Trichodermene A, isolated from *T. longibrachiatum*, demonstrated significant antifungal activity against two strains of anthracnose (*Colletotrichum lagrnarium* and *C. fragariae*) and *B. cinerea*, with minimum inhibitory concentration (MIC) values ranging from 8 to 64 μg/mL [41]. Song isolated the cyclonerane sesquiterpene 11-methoxy-9-cycloneren-3,7-diol from *T. harzianum* X-5, which significantly inhibited the growth of *Chattonella marina* and *Karlodinium veneficum*, with IC_50_ values of 0.66 μg/mL and 2.2 μg/mL, respectively [42]. *T. hamatum* FB10 has been reported to exhibit antagonistic activity against pathogenic fungi, including *Sclerotinia sclerotiorum*, *Rhizoctonia solani*, *Alternaria radicina*, *Alternaria citri*, and *Alternaria dauci*, through the production of biologically active volatile secondary metabolites [43]. Two new cyclonerane sesquiterpenes, 5-hydroxyepicyclonerodiol oxide and 4-hydroxyepicyclonerodiol oxide, along with a novel natural product, trichodermol chlorohydrin, were isolated from *T. hamatum* Z36-7. These compounds exhibit growth inhibitory effects on a broad spectrum of bacteria and phytoplankton [44]. Current research indicates that *T. hamatum* has potential for biocontrol; however, further studies on its secondary metabolites for biocontrol applications are still needed.

In the present study, we isolated and purified secondary metabolites extracted using ethyl acetate from *T. hamatum* and identified the structures of the previously undescribed compounds **1**–**7**, as well as the known compound **8**, utilizing HRESIMS, NMR, UV, IR, circular dichroism, and Mosher’s method, in conjunction with computational chemistry. Compounds **1**, **2**, and **9** are harziane diterpenes characterized by a unique 6-5-4-7 tetracyclic carbon skeleton, compounds **3**–**8** are cyclonerane sesquiterpenes, among which compound **5** represents the first -OOH-substituted cyclonerane sesquiterpene to be discovered, while compound **10** was identified as dehydroacetic acid.

Dehydroacetic acid (**10**) was first isolated from *Solandra nitida* in 1866 and has been utilized as a food preservative due to its efficacy in inhibiting the growth of molds, yeasts, and bacteria. Its derivatives have been investigated for the development of effective antimicrobial agents against various bacteria and fungi [45]; however, research on its antagonistic effects against agricultural pathogens remains relatively limited. In this study, the fungicidal activities of compounds **1**–**9** and dehydroacetic acid (**10**) against six agricultural pathogens were assessed. The results indicated that some of these compounds exhibited promising fungicidal activities against pathogenic fungi. Notably, dehydroacetic acid (**10**) significantly inhibited the growth of both *B. cinerea* and *R. solani*. Furthermore, in vivo examinations of the effects of dehydroacetic acid (**10**) on the control of *B. cinerea* and *S. sclerotiorum* suggested its potential for development as an agricultural antibiotic. Additionally, both the research of Baazeem [43] and the activity tests in the present study showed that *T. hamatum* exhibited antagonistic effects against *S. sclerotiorum* and *R. solani*, suggesting that dehydroacetic acid (**10**) may be an effective substance for the antagonistic effect of *T. hamatum*, which was also evidenced by the substantial number of isolations of dehydroacetic acid (**10**).

This work not only enriched the active material base of *T. hamatum*, but also revealed the active components for the biocontrol role of *T. hamatum* and advanced the application of its secondary metabolites in controlling agricultural pathogenic fungi, thereby providing a valuable reference for the development of fungal biocontrol strategies.

## 5. Patents

Chengdu University of Traditional Chinese Medicine has filed patents ZL202311414683.0 and ZL202311251991.6 concerning the in vitro antifungal activity of compounds **2** and **3**.

## Figures and Tables

**Figure 1 jof-10-00755-f001:**
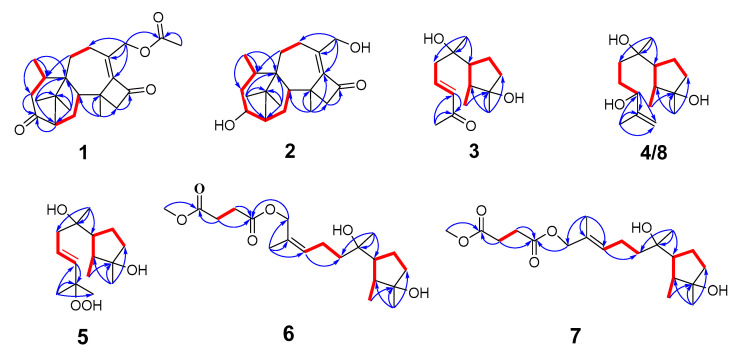
Key HMBC (bold lines) and ^1^H–^1^H COSY (arrows) correlations of compounds **1**–**8**.

**Figure 2 jof-10-00755-f002:**
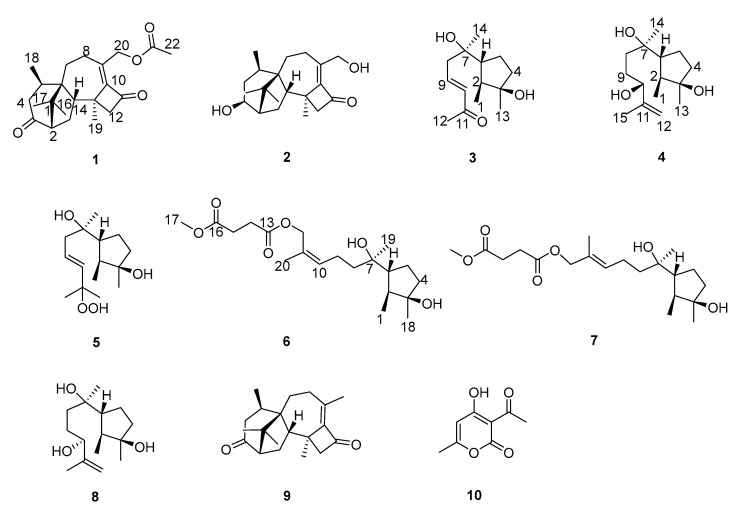
Chemical structures of compounds **1**–**10**.

**Figure 3 jof-10-00755-f003:**
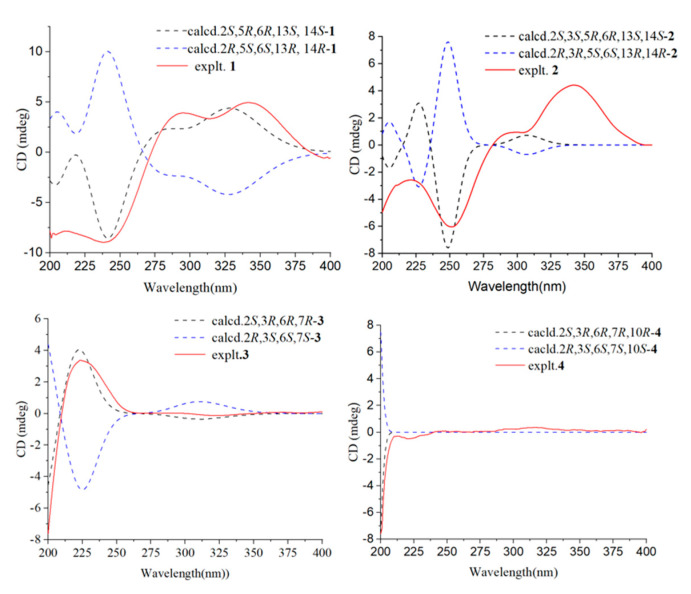
Calculated ECD spectra and experimental ECD curves of compounds **1**–**4** in MeOH.

**Figure 4 jof-10-00755-f004:**
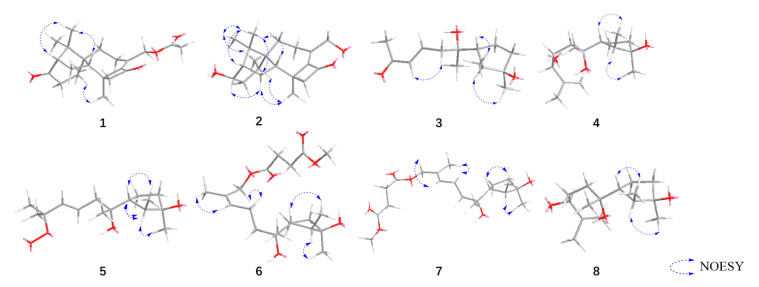
Key NOESY correlations of compounds **1**–**8**.

**Figure 5 jof-10-00755-f005:**
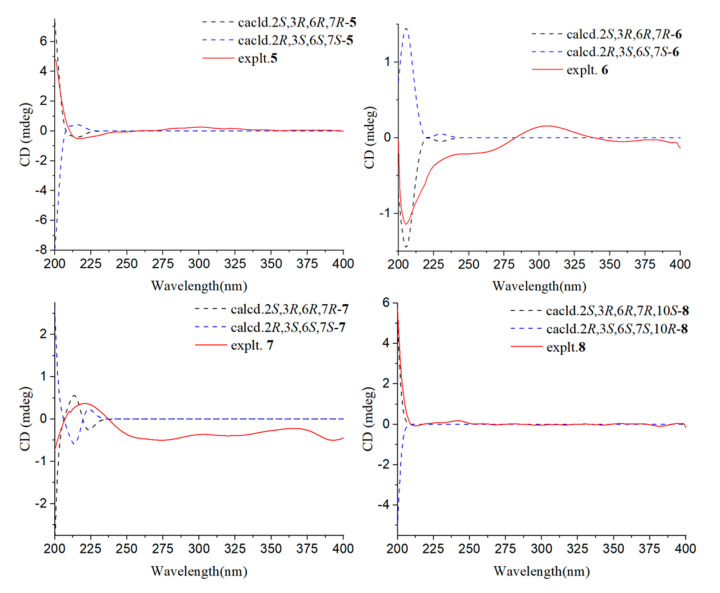
Calculated ECD spectra and experimental ECD curves of compounds **5**–**8** in MeOH.

**Figure 6 jof-10-00755-f006:**
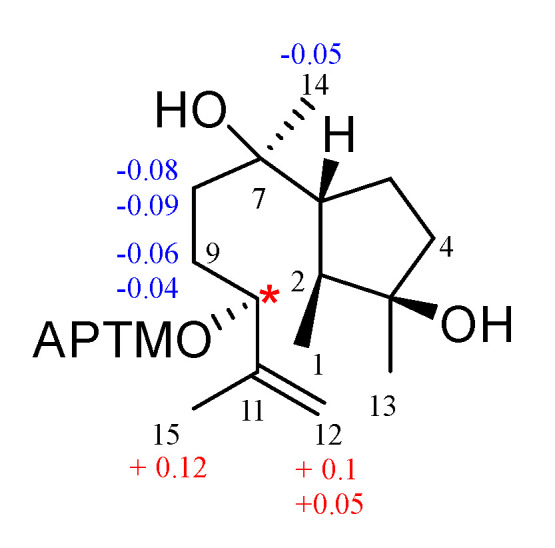
Δ*δ_S-R_* value (ppm) of the MTPA ester of **8**.

**Figure 7 jof-10-00755-f007:**
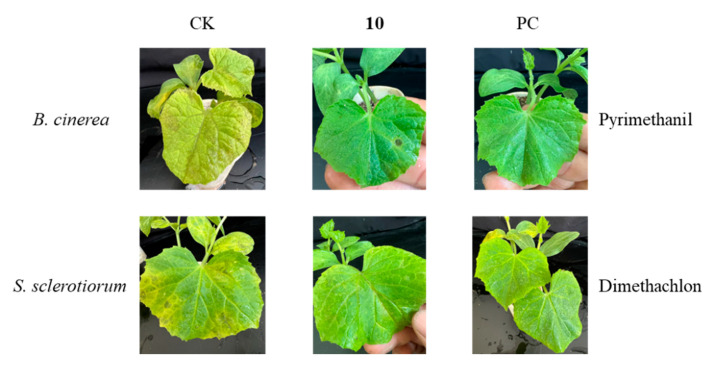
Activities of compound **10** against *B. cinerea* and *S. sclerotiorum* in vivo (CK: blank control; PC: positive control).

**Table 1 jof-10-00755-t001:** NMR data for compounds **1**–**2** and **6**–**7** (600 MHz, CDCl_3_).

Pos.	1	2	6	7
	*δ*_C_*,* Type	*δ*_C_*,* Type	*δ*_C_*,* Type	*δ*_C_*,* Type
1	49.7, C	45.8, C	14.7, CH_3_	14.7, CH_3_
2	59.4, CH	49.6, CH	44.4, CH	44.4, CH
3	214.2, C	74.4, CH	81.4, C	81.5, C
4	42.7, CH_2_	34.3, CH_2_	40.5, CH_2_	40.5, CH_2_
5	30.1, CH	28.2, CH	24.5, CH_2_	24.5, CH_2_
6	51.8, C	50.4, C	54.6, CH	54.4, CH
7	30.3, CH_2_	30.4, CH_2_	74.8, C	74.9, C
8	23.8, CH_2_	24.6, CH_2_	40.5, CH_2_	40.0, CH_2_
9	141.1, C	154.0, C	22.5, CH_2_	22.5, CH_2_
10	153.0, C	148.9, C	131.2, CH	129.9, CH
11	196.5, C	200.2, C	129.9, C	130.2, C
12	60.2, CH_2_	58.7, CH_2_	63.6, CH_2_	70.6, CH_2_
13	40.4, C	40.4, C	172.4, C	172.3, C
14	52.9, CH	51.4, CH	29.1, CH_2_	29.1, CH_2_
15	26.7, CH_2_	27.6, CH_2_	29.3, CH_2_	29.3, CH_2_
16	25.2, CH_3_	26.8, CH_3_	172.9, C	172.9, C
17	23.4, CH_3_	23.6, CH_3_	52.0, CH_3_	52.0, CH_3_
18	21.1, CH_3_	21.4, CH_3_	26.2, CH_3_	26.2, CH_3_
19	20.6, CH_3_	21.6, CH_3_	25.0, CH_3_	25.1, CH_3_
20	63.3, CH_2_	67.3, CH_2_	21.6, CH_3_	21.6, CH_3_
21	170.9, C			
22	21.0, CH_3_			

**Table 2 jof-10-00755-t002:** ^1^H-NMR data for compounds **1**–**2** and **6**–**7** (150 MHz, CDCl_3_).

Pos.	1	2	6	7
	*δ*_H_ (*J* in Hz)	*δ*_H_ (*J* in Hz)	*δ*_H_ (*J* in Hz)	*δ*_H_ (*J* in Hz)
1			1.04, d (6.8)	1.05, d (6.8)
2	2.29, m	1.84, dd (8.2, 3.7)	1.60, m	1.61, m
3		3.98, dd (3.6, 6.6)		
4a	2.90, m	2.42, d (16.9)	1.68, m	1.69, m, 1H
4b	2.09, m	1.50, d (15.3)	1.56, m	1.57, m
5a	2.90, m	2.45, m	1.85, m	1.86, m
5b			1.55, m	1.55, m
6			1.84, m	1.85, m
7a	1.97, m	1.97, m		
7b	1.40, m	1.25, m		
8a	2.29, m	2.40, m	1.49, m	1.51, t (8.4)
8b		2.00, m		
9a			2.19, m	2.12, m
9b			2.12, m	
10			5.41, t (6.7)	5.47, td (7.2, 1.4)
12a	2.66, d (16.6)	2.57, d (16.7)	4.66, d (11.9)	4.48, s
12b	2.51, d (16.4)	2.46, d (16.9)	4.60, d (11.9)	
14	2.52, m	2.14, dd (11.3, 8.9)	2.64, m	2.65, m
15a	2.05, m	1.90, m	2.64, m	2.65, m
15b	1.54, m	1.09, dd (14.0, 9.3)		
16	1.00, s	0.87, s		
17	1.01, s	1.33, s	3.69, s	3.69, s
18	1.13, d (7.2)	1.18, d (7.6)	1.26, s	1.26, s
19	1.54, s	1.51, s	1.15, s	1.17, s
20a	5.12, d (12.8)	4.40, d (18)	1.74, d (1.5)	1.66, d (1.5)
20b	4.76, d (12.9)	4.20, d (18.2)		
22	2.10, s			

**Table 3 jof-10-00755-t003:** NMR data for compounds **3**–**5** and **8** (600 MHz, CDCl_3_).

Pos.	3	4	5	8
	*δ*_C_*,* Type	*δ*_C_*,* Type	*δ*_C_*,* Type	*δ*_C_*,* Type
1	14.6, CH_3_	14.7, CH_3_	14.5, CH_3_	14.6, CH_3_
2	44.6, CH	44.5, CH	44.6, CH	44.5, CH
3	81.4, C	81.5, C	81.6, C	81.4, C
4	40.4, CH_2_	40.5, CH_2_	40.4, CH_2_	40.5, CH_2_
5	24.6, CH_2_	24.5, CH_2_	24.5, CH_2_	24.5, CH_2_
6	54.9, CH	54.8, CH	54.4, CH	54.8, CH
7	75.0, C	74.7, C	74.8, C	74.8, C
8	43.8, CH_2_	36.6, CH_2_	43.6, CH_2_	35.8, CH_2_
9	144.1, CH	29.4, CH_2_	126.7, CH_2_	29.1, CH_2_
10	134.2, CH	76.7, CH	138.1, CH	76.0, CH
11	198.5, C	147.7, C	82.0, C	147.7, C
12	27.2, CH_3_	111.2, CH_2_	24.6, CH_3_	111.0, CH_2_
13	26.2, CH_3_	26.2, CH_3_	26.1, CH_3_	26.2, CH_3_
14	25.9, CH_3_	25.2, CH_3_	25.2, CH_3_	25.2, CH_3_
15		17.8, CH_3_	24.2, CH_3_	18.2, CH_3_

**Table 4 jof-10-00755-t004:** ^1^H-NMR data for compounds **3**–**5** and **8** (150 MHz, CDCl_3_).

Pos.	3	4	5	8
	*δ*_H_ (*J* in Hz)	*δ*_H_ (*J* in Hz)	*δ*_H_ (*J* in Hz)	*δ*_H_ (*J* in Hz)
1	1.06, d (6.8)	1.05, d 6.7)	1.04, d (6.7)	1.04, d (6.9)
2	1.62, m	1.62, m	1.60, m	1.59, m
4a	1.72, m	1.69, m	1.68, m	1.69, m
4b	1.58, m	1.57, m	1.56, m	1.57, m
5a	1.91, m	1.86, m	1.86, m	1.87, m
5b	1.58, m	1.56, m	1.57, m	1.56, m
6	1.86, m	1.86, m	1.86, m	1.87, m
8a	2.45, dd (14.8, 7.6)	1.60, m	2.23, m	1.57, m
8b	2.36, dd (13.9, 8.2)	1.48, m		1.50, m
9a	6.88, m	1.65, m	5.74, dt (15.0, 7.4)	1.71, m
9b				1.61, m
10	6.13, d (16.0)	4.05, dd (7.2, 5.3)	5.63, d (15.8)	4.09, dd (7.6, 4.6)
12a	2.27, s	4.95, s	1.32, s	4.97, s
12b		4.84, s		4.86, s
13	1.27, s	1.26, s	1.25, s	1.26, s
14	1.19, s	1.16, s	1.14, s	1.16, s
15		1.74, s	1.33, s	1.72, s

**Table 5 jof-10-00755-t005:** Activities of compounds **1**–**10** against six kinds of pathogenic fungi ^a^.

Compound	Fungicidal Activities (%) at 50 μg/mL					
	*A.S*.	*F.G*.	*P.C.*	*S.S*.	*B.C.*	*R.S*.
**1**	52.2 ± 2.3	56.5 ± 4.2	45.7 ± 1.5	**85.9 ± 3.7**	44.7 ± 5.1	19.3 ± 0.6
**2**	39.1 ± 3.4	17.4 ± 1.1	21.7 ± 2.6	**73.1 ± 2.6**	26.3 ± 2.5	46.5 ± 4.5
**3**	56.5 ± 4.2	34.8 ± 2.0	42.8 ± 2.5	**64.1 ± 2.5**	**81.6 ± 3.4**	28.1 ± 3.4
**4**	47.8 ± 1.7	23.9 ± 1.6	32.6 ± 2.4	**73.1 ± 4.2**	42.1 ± 2.6	15.1 ± 2.3
**5**	45.8 ± 2.4	26.1 ± 2.5	23.9 ± 2.2	**67.9 ± 3.4**	39.5 ± 4.3	19.3 ± 3,6
**6**	30.4 ± 3.5	**84.8 ± 3.1**	26.1 ± 1.1	**75.6 ± 1.8**	47.4 ± 2.1	17.4 ± 2.1
**7**	34.8 ± 1.9	15.2 ± 1.4	35.0 ± 3.1	51.3 ± 2.3	34.2 ± 1.4	14.0 ± 1.5
**8**	30.4 ± 2.4	28.3 ± 2.6	37.0 ± 2.2	53.8 ± 4.1	26.3 ± 2.6	51.2 ± 3.9
**9**	12.5 ± 1.2	23.1 ± 3.4	9.4 ± 0.6	**76.1 ± 3.5**	46.2 ± 3.6	59.4 ± 4.3
**10**	56.3 ± 3.6	38.5 ± 3.2	28.1 ± 1.8	**100.0 ± 0.0**	**100.0 ± 0.0**	**82.8 ± 2.5**

^a^ *A.S.*: *Alternaria solani*; *F.G.*: *Fusarium graminearum*; *P.C.*: *Phytophthora capsici*; *S.S.*: *Sclerotinia sclerotiorum*; *B.C.*: *Botrytis cinerea*; *R.S.*: *Rhizoctonia solani*.

**Table 6 jof-10-00755-t006:** Activities of compound **10** against *Botrytis cinerea* and *Sclerotinia sclerotiorum* in vivo.

Preventative Efficiency (%) ^a^		
Pathogenic Fungi	*Botrytis Cinerea*	*Sclerotinia Sclerotiorum*
**10**	65.8	49.1
PC ^b^	88.6	100

^a^ ANOVA was analyzed using Duncan’s new multiple range test. ^b^ PC: positive control.

## Data Availability

The authors will make all raw data supporting the conclusions of this article available to any qualified researcher, without undue reservation.

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
