# Peer review of "Structure and Fungicidal Activity of Secondary Metabolites Isolated from Trichoderma hamatum b-3"

_jof, 2024, doi:10.3390/jof10110755_

Round 1

Reviewer 1 Report

This manuscript discusses the secondary metabolites from Trichoderma hamatum b-3. In vitro fungicidal activity showed that some compounds had great inhibitory activity against pathogenic fungi is a popular theme. In this aspect, the study is important and relevant, and its results may provide some practical application of the studied strain in the future. However, the description of the introduction and discussion in the study, and title should be improved before the manuscript acceptance (see comments below).

First, please correct the title of the work because it suggests auto-plagiarism - Secondary metabolites isolated from Trichoderma hamatum b-3 and their fungicidal activity - https://www.sciencedirect.com/science/article/abs/pii/S0367326X24000637.

The introduction is extremely short and requires more detailed explanations.

-Explain in more detail biological control and its importance. Line 34-38.

- The significance of Trichoderma is not highlighted, nor are the plant pathogens it affects. Add appropriate references.

-Discussion: The discussion is generally written, nowhere are similar secondary metabolites of Trichoderma fungi and their activity mentioned. There is no comparison within the genus itself. It needs to be expanded with adequate examples and supplemented with references.

Author Response

  1. First, please correct the title of the work because it suggests auto-plagiarism - Secondary metabolites isolated from Trichoderma hamatum b-3 and their fungicidal activity https://www.sciencedirect.com/science/article/abs/pii/S0367326X24000637.

Response: Thank you for your thoughtful suggestion. We have revised the title to "Structure and Fungicidal Activity of Secondary Metabolites Isolated from Trichoderma hamatum b-3" based on your careful review.

  1. The introduction is extremely short and requires more detailed explanations.

- Explain in more detail biological control and its importance. Line 34-38.

Response: Thank you for your careful review, we have included details regarding the significance of biological control in lines 38-46.

- The significance of Trichoderma is not highlighted, nor are the plant pathogens it affects. Add appropriate references.

Response: we have supplemented this section in lines 47-50 and 55-58 based on your kind suggestion.

-Discussion: The discussion is generally written, nowhere are similar secondary metabolites of Trichoderma fungi and their activity mentioned. There is no comparison within the genus itself.

Response: Thank you for your kind suggestions, we have amended lines 417-431 in response to your suggestion.

  1. It needs to be expanded with adequate examples and supplemented with references.

Response: Thanks for your suggestion and we have added some examples and references in lines 417-431 and 451-457.

Reviewer 2 Report

1. Overall, the abstract provides a good summary of the research and its outcomes. However, it would benefit from more context on the novelty of the findings and a clearer articulation of the broader impact of the research. the biological or chemical mechanisms that might explain the effectiveness of compound 10. Please modified the abstract.

2. Figure 1 need molecular weight

3. In section 2.2 collected from Mount Emei and identified to be T. hamatum by Shanghai… how was collected and identified please write in detail

4. In section 2.3 The cultured fermentation broth was centrifuged at 4000 r/min at a high speed. for how many times write detail

5. Please modified figure 4 increase the size

1. Overall, the abstract provides a good summary of the research and its outcomes. However, it would benefit from more context on the novelty of the findings and a clearer articulation of the broader impact of the research. the biological or chemical mechanisms that might explain the effectiveness of compound 10. Please modified the abstract.

2. Figure 1 need molecular weight

3. In section 2.2 collected from Mount Emei and identified to be T. hamatum by Shanghai… how was collected and identified please write in detail

4. In section 2.3 The cultured fermentation broth was centrifuged at 4000 r/min at a high speed. for how many times write detail

5. Please modified figure 4 increase the size

Author Response

Overall, the abstract provides a good summary of the research and its outcomes. However, it would benefit from more context on the novelty of the findings and a clearer articulation of the broader impact of the research. the biological or chemical mechanisms that might explain the effectiveness of compound 10. Please modified the abstract.

Response: Thank you for your thoughtful suggestions, we've supplemented the abstract section in lines 25-28.

  1. Figure 1 need molecular weight

Response: Thank you for your suggestion. However, it is customary to label it solely with the compound number, omitting the molecular weight (10.3390/jof10090611, 10.3390/jof10090616).

  1. In section 2.2 collected from Mount Emei and identified to be T. hamatum by Shanghai… how was collected and identified please write in detail

Response: Thanks to your kind suggestion, we've added this section to lines 95-115 in the manuscript.

  1. In section 2.3 The cultured fermentation broth was centrifuged at 4000 r/min at a high speed. for how many times write detail

Response: Based on your suggestion, we have added the number of centrifuges in lines 130-131 of the manuscript.

  1. Please modified figure 4 increase the size

Response: Thank you for your suggestion, we've amended figure 4 in lines 309 and 382.

Reviewer 3 Report

The manuscript is devoted to the search for new secondary metabolites of fungi of the genus Trichoderma that have fungicidal activity. However, most of the manuscript is devoted to the isolation and determination of the chemical structure of these compounds. Little attention is paid to experiments to test their fungicidal activity; they were carried out only on one plant at one concentration. In my opinion, the manuscript does not comply with JOF; it is more suitable for a chemical journal. Nevertheless, I have reviewed the manuscript.

1. In the introduction, the authors report that they have previously isolated several new compounds with bioactive properties from the fungi Chaetomium elatum and Nigrospora sphaerica. Has this led to the development of biofungicides based on these compounds?

2. It should be indicated which plant diseases are caused by Alternaria solani, Fusarium graminearum, Phytophthora capsici, Sclerotinia sclerotiorum, Botrytis cinerea and Rhizoctonia solani. Why was a concentration of 50 μg/mL chosen to test the fungicidal activity of compounds 1-10 in vitro? Why didn't the authors use multiple concentrations?

3. Lines 184-185. "Both compounds, along with the controls, were dissolved in N, N-dimethylformamide and subsequently diluted to a concentration of 200 μg/mL." What are these two compounds?

4. Why was it necessary to use the solvent N, N-dimethylformamide? Why couldn't the compound be dissolved in water immediately?

5. Why was a concentration of 200 μg/mL chosen to test the fungicidal activity of compound 10 in vivo, which is 4 times higher than the concentration used in the in vitro experiment? Why didn't the authors use multiple concentrations?

6. Why were pyrimethanil and dimethaclone chosen as standards?

7. Figure 6 shows the symbols СК and РС. What do they mean? Table 6 also contains the symbol РС, but no СК.

8. The manuscript lacks the sections Statistical Analysis and Conclusion.

9. Discussion is too short.

Author Response

The manuscript is devoted to the search for new secondary metabolites of fungi of the genus Trichoderma that have fungicidal activity. However, most of the manuscript is devoted to the isolation and determination of the chemical structure of these compounds. Little attention is paid to experiments to test their fungicidal activity; they were carried out only on one plant at one concentration. In my opinion, the manuscript does not comply with JOF; it is more suitable for a chemical journal. Nevertheless, I have reviewed the manuscript.

Response: Thank you for your kind advice. Fungal secondary metabolites play a crucial role in the study of fungi. Given that numerous similar articles have already been published in Journal of Fungi (10.3390/jof10090611, 10.3390/jof10090616), we are submitting our article to the journal.

  1. In the introduction, the authors report that they have previously isolated several new compounds with bioactive properties from the fungi Chaetomium elatum and Nigrospora sphaerica. Has this led to the development of biofungicides based on these compounds?

Response: Thank you for your question. We have been investigating the natural products of microbes for many years, and previous studies have reported numerous secondary metabolites, some of which exhibit significant anti-inflammatory activity. However, none of these metabolites have undergone screening for fungicidal activity. It is only in the past two years that we have begun to focus on their fungicidal properties.

  1. It should be indicated which plant diseases are caused by Alternaria solani, Fusarium graminearum, Phytophthora capsici, Sclerotinia sclerotiorum, Botrytis cinerea and Rhizoctonia solani. Why was a concentration of 50 μg/mL chosen to test the fungicidal activity of compounds 1-10 in vitro? Why didn't the authors use multiple concentrations?

Response: Thank you for your careful review. Alternaria solani is responsible for causing tomato early blight, Fusarium graminearum leads to wheat blast, Phytophthora capsici is associated with chilli mildew, Sclerotinia sclerotiorum causes rape sclerotinia stem rot, and Botrytis cinerea is known to cause grey mould in cucumbers and tomatoes. Additionally, Rhizoctonia solani is implicated in rice blight. We have added to lines 202-205 of the text.

We selected a concentration of 50 μg/mL for our tests based on the initial common concentration of new compounds recommended for in vivo general screening by the National Pesticide Engineering Research Center (Tianjin, China). Following the confirmation of satisfactory activity at this concentration, we proceeded with further in vivo potting tests.

  1. Lines 184-185. "Both compounds, along with the controls, were dissolved in N, N-dimethylformamide and subsequently diluted to a concentration of 200 μg/mL." What are these two compounds?

Response: Sorry for the typo, it should be that the compound and control samples are dissolved separately in N, N-dimethylformamide. We've fixed the error on line 126.

  1. Why was it necessary to use the solvent N, N-dimethylformamide? Why couldn't the compound be dissolved in water immediately?

Response: Thank you for your careful review. This series of compounds is not water-soluble and cannot be dissolved directly in water; instead, they require the addition of organic solvents for dissolution. N, N-dimethylformamide is one such organic solvent, and its use is a standardized procedure at the National Pesticide Engineering Research Center in Tianjin, China.

  1. Why was a concentration of 200 μg/mL chosen to test the fungicidal activity of compound 10 in vivo, which is 4 times higher than the concentration used in the in vitro experiment? Why didn't the authors use multiple concentrations?

Response: Thank you for your careful review. The initial screening concentration for the in vivo pot test was set at 200 μg/mL. Multiple concentrations would only be tested if this concentration proved to be effective.

  1. Why were pyrimethanil and dimethaclone chosen as standards?

Response: Thank you for your thoughtful questions. Pyrimethanil is the primary fungicide utilized in agricultural production for the control of grey mould in cucumbers, while dimethaclone is a widely used agent for managing sclerotinia disease in rapeseed.

  1. Figure 6 shows the symbols СK and РС. What do they mean? Table 6 also contains the symbol РС, but no CK.

Response: CK served as a blank control, in which only treatments containing the same concentration of organic solvent as each experimental treatment were applied. PC represented the positive control, with Pyrimethanil used for managing cucumber grey mould and dimethaclone employed as the positive control for rapeseed sclerotinia disease. This abbreviation has been noted on lines 216 and 218.

  1. The manuscript lacks the sections Statistical Analysis and Conclusion.

Response: Thank you for your kind suggestion. The experiments described in section 3.2 involve general pathogen target screening based on established criteria for evaluating new compounds. This process primarily assesses inhibition rates and control profiles without conducting significance analyses, reserving such analyses for rescreening or field trials. Regarding the statistical methodology for the live potting experiment, we have incorporated the necessary details in line 410. In light of the journal's requirement to combine the conclusion and discussion sections into a single section, we have merged the conclusion into the discussion. You can see the above changes in lines 417-431 and 451-457.

  1. Discussion is too short.

Response: Based on your suggestions, we have added and modified the discussion section in lines 417-431 and 451-457.

Round 2

Reviewer 3 Report

Translator        

Translator        

Thank you for your answers.

Translator        

Translator        

Thank you for your answers.